# Ten Considerations for Integrating Patient-Reported Outcomes into Clinical Care for Childhood Cancer Survivors

**DOI:** 10.3390/cancers15041024

**Published:** 2023-02-06

**Authors:** Madeline R. Horan, Jin-ah Sim, Kevin R. Krull, Kirsten K. Ness, Yutaka Yasui, Leslie L. Robison, Melissa M. Hudson, Justin N. Baker, I-Chan Huang

**Affiliations:** 1Department of Epidemiology and Cancer Control, St. Jude Children’s Research Hospital, Memphis, TN 38105, USA; 2School of AI Convergence, Hallym University, Chuncheon 24252, Republic of Korea; 3Department of Psychology, St. Jude Children’s Research Hospital, Memphis, TN 38105, USA; 4Department of Oncology, St. Jude Children’s Research Hospital, Memphis, TN 38105, USA

**Keywords:** childhood cancer survivors, patient-reported outcomes, health-related quality of life, implementation, symptoms

## Abstract

**Simple Summary:**

Patient-reported outcome measures (PROMs) are a useful way to assess the subjective experiences of health-related quality of life, functional status, symptoms, and other outcomes in childhood cancer survivors. In survivorship care, PROMs can be used to monitor health status and inform medical decision making. This article provides 10 important considerations for clinicians when they are assessing patient-reported outcomes for childhood cancer survivors. From choosing the right measure to selecting a strategy for clinical response, the purpose of these considerations is to support clinicians in implementing PROMs in their practice while keeping in mind some of the practical barriers and solutions of using PROMs with childhood cancer survivors. We end with an example of a framework for integrating PROMs into the clinical workflow that uses cutting-edge technologies (e.g., mHealth, natural language processing, and machine learning) to minimize interruptions to the clinical workflow and maximize the powerful utility of PROMs in cancer survivorship care.

**Abstract:**

Patient-reported outcome measures (PROMs) are subjective assessments of health status or health-related quality of life. In childhood cancer survivors, PROMs can be used to evaluate the adverse effects of cancer treatment and guide cancer survivorship care. However, there are barriers to integrating PROMs into clinical practice, such as constraints in clinical validity, meaningful interpretation, and technology-enabled administration of the measures. This article discusses these barriers and proposes 10 important considerations for appropriate PROM integration into clinical care for choosing the right measure (considering the purpose of using a PROM, health profile vs. health preference approaches, measurement properties), ensuring survivors complete the PROMs (data collection method, data collection frequency, survivor capacity, self- vs. proxy reports), interpreting the results (scoring methods, clinical meaning and interpretability), and selecting a strategy for clinical response (integration into the clinical workflow). An example framework for integrating novel patient-reported outcome (PRO) data collection into the clinical workflow for childhood cancer survivorship care is also discussed. As we continuously improve the clinical validity of PROMs and address implementation barriers, routine PRO assessment and monitoring in pediatric cancer survivorship offer opportunities to facilitate clinical decision making and improve the quality of survivorship care.

## 1. Introduction

The global 5-year survival rate of childhood cancer is 37%, ranging from 8% in eastern Africa to 83% in North America [1]. Childhood cancer survivors frequently develop organ-based dysfunction or progressive chronic health conditions (CHCs) [2,3,4], which often coincide with impaired health-related quality of life (HRQOL) compared to siblings and the general population [5,6]. These subjective health outcomes can be assessed as patient-reported outcomes (PROs).

PROs capture patient experiences including physical, somatic, and psychological symptoms, functional status, overall quality of life (QOL), and subjective well-being [7]. PRO measures (PROMs) have been used with childhood cancer survivors to assess varying aspects (e.g., presence, frequency, severity, interference) of symptoms, monitor health status, and inform medical decision making. Up to 75% of pediatric cancer survivors have at least two concurrent symptoms [8], which significantly impacts overall QOL [9,10] and adverse health outcomes [11,12,13] more than individual symptoms. As more pediatric cancer patients become long-term survivors, worsening symptom burden can well predict the progression of various CHCs in adulthood [14].

Most PROMs have been developed for pediatric cancer patients; however, less attention has been given to PROMs that are developed or validated specifically for survivors. While most clinicians and patients recognize the value of using PROMs in survivorship care to improve patient–provider communication, focus on issues important to the survivor, and provide needed referrals [15,16,17], there are also barriers to integrating PROMs into practice. Clinicians have reported that the most common barriers include limited time and manpower for assessing PROs, limited training on PROM administration, lack of clinically meaningful cutpoints for scores, limited skills related to scoring and interpreting results, lack of recommendations for follow-up and referral services, unavailability of computerized mode for PROM administration, and a lack of evidence that PRO assessment improves care [16,18,19].

There is a gap between the benefit of administrating PROMs for cancer survivors and the challenges of integrating PROMs into clinical practice. In light of this gap, users’ guidelines to integrate PROMs into practice have been developed by the International Society for Quality of Life Research [20] and the Patient-Centered Outcomes Research Institute [21]. These documents are useful for guiding clinicians (i.e., anyone who provides care for childhood cancer survivors) in the process of integrating PROMs. There are, however, no one-size-fits-all PROMs for all childhood cancer survivors because survivors receive different anticancer therapies at different ages and developmental stages and develop varying late effects that require using clinically sensitive or meaningful PROMs to inform clinical decision making. This article provides 10 clinically actionable considerations for clinicians using PROMs for selecting the right measures, ensuring survivors complete the measures, interpreting the results, and selecting a strategy for clinical response (see Table 1). Clinicians may reference these considerations to effectively implement appropriate PROMs into clinical care for survivors. We also extend the last consideration with an example of integrating novel PRO data collection into the clinical workflow for childhood cancer survivorship care to minimize interruptions to the clinical workflow and maximize the powerful utility of PROMs in clinical cancer survivorship care.

**Table 1 cancers-15-01024-t001:** A summary of ten considerations for primary care physicians/oncologists when implementing PRO measures for pediatric cancer population.

**Choosing the right measure** Consideration 1: Purpose of using a PROM Is this measure used for classification (e.g., based on symptom severity)? Is this measure used for prediction (e.g., symptom onset, late effects, survival)? Is this measure used for communication (e.g., as a starting point for a conversation with a survivor)? Consideration 2: Health profile vs. health preference approaches Health profile measures Advantage: useful for examining the impact of disease or treatment on specific PRO domains; suitable for making care decisions for individuals Disadvantage: scale values and summary scores may be unfamiliar or difficult to interpret Health preference measures Advantage: useful for making decisions regarding the distribution of health care resources across different diseases; common metric (0 = dead, 1 = optimal health) Disadvantage: preference weights developed from the general population in Commonwealth countries and may not be applicable for all countries; some scaling methods are difficult for children to complete Consideration 3: Measurement properties Standard measurement properties and definitions (see a full list in Table 2) Clinical relevance and actionability: content validity (measuring constructs that are important and/or relevant to the clinician and survivor); responsiveness to change (PROM’s sensitivity to changes in underlying health conditions); response shift (measuring genuine outcomes if the survivor has altered their perception of the PROs) **Ensuring participants complete the measure** Consideration 4: Data collection method Paper-and-pencil Advantage: easily accessible; does not require technology Disadvantage: requires more manpower to score each assessment Computerized adaptive tests (CAT) on a computer Advantage: less burdensome for survivors; more time-efficient; provides higher measurement precision Disadvantage: requires technology; not all PROMs available in CAT format Web-based Advantage: available for anyone with Internet access; may be preferable to a mobile device application for older adults Disadvantage: requires technology; requires Internet access Mobile devices (e.g., tablet, smart phone) Advantage: allows for notification reminders for survivors to complete PROMs; allows for real-time assessment Disadvantage: requires technology; inaccessible for people without a device; may lead to burnout if survivors are asked to complete PROMs too often Consideration 5: Data collection frequency and longitudinal assessment Momentarily (real time)—for conditions that require intensive monitoring Daily—for conditions that require intensive monitoring Weekly—currently common practice Specific timepoints—per treatment protocol During clinical visits—time intervals between assessments will be uneven Data collection throughout the survivorship journey—from childhood to adulthood Consideration 6: Survivor capacity Age—the majority of measures are appropriate for ages ≥ 8 years; < 8 years may use a visual scale (e.g., Child Health and Illness Profile (CHIP), Faces Pain Scale—Revised (FPS-R)) or proxy report Developmental or health impairment—limited reading level, cognitive functional deficits, and vision or hearing problems that require special assistance or measurement systems Consideration 7: Self- vs. proxy reports Ideally self-reports Proxy reports if age, cognitive ability, or health impairment inhibit self-reporting Consider how to integrate self- with proxy-reported data **Interpreting the results** Consideration 8: Scoring methods Unweighted methods: summation; mean scores Advantage: easy to calculate Disadvantage: may not accurately reflect outcome differences Weighted methods: confirmatory factor analysis (CFA); item response theory (IRT) Advantage: more sensitive to changes in health conditions over time Disadvantage: more difficult to calculate; not as familiar to most clinicians Consideration 9: Clinical meaning and interpretability Meaningful cutpoints—for deciding the severity of PROs Minimally important differences (MIDs)—for deciding the minimum threshold for clinical action **Selecting a strategy for clinical response** Consideration 10: Integration into the clinical workflow Integration with electronic health records (EHRs) Augmentation with artificial-intelligence (AI)-driven analytics and decision support tools for PROs Triggering patient–doctor communication between clinic visits Tracking longitudinal PROs

**Table 2 cancers-15-01024-t002:** Measurement properties: Importance for clinical consideration.

Measurement Property	Importance for Clinical Consideration
Content validity	Demonstrates relevance for the construct to be measured in the context of pediatric cancer survivors.
Internal consistency	Demonstrates the unidimensionality of each subscale separately.
Reliability	Demonstrates the items that appropriately capture the same concept of a given PRO at the same time or over different time points.
Measurement error	Quantifies unobserved components of PROMs that influence the accuracy of the assessment.
Construct validity/known-group validity	Demonstrates how PROMs are sensitive to levels of clinical anchors or parameters relevant to survivors.
Structural validity	Demonstrates the appropriateness of PROMs based on a reflective theoretical model, not a formative model.
Cross-cultural validity/measurement invariance	Demonstrates how the constructs of PROMs are comparable across different demographic or cultural subgroups, including those who differ in ethnicity, language, gender, age, or patient population.
Criterion validity	Demonstrates the degree to which the PROM agrees with a gold standard. Only necessary if a gold standard measure of the construct of interest in pediatric cancer survivors is available. There are very few gold standard PROMs for pediatric cancer survivors.
Responsiveness	Demonstrates how the change in PRO scores reflects a change in clinical status in a way that is meaningful to the patient or clinician.
Interpretability	Demonstrates how survivors can comprehend the meaning of PRO items.
Feasibility	Decreases the burden of PRO assessment, especially among survivors who are ill or have poor functional status.
Predictive validity	Facilitates predicting adverse events (e.g., the onset of late effect) or premature death for high-risk survivor populations.
Cutpoints/MID	Uses an anchor-based approach with multiple anchors in calculating the MID or cutpoint that can be used to help clinicians interpret PRO scores for clinical decision making.
Response shift	Calculates response-shift-adjusted PRO scores to reflect genuine PRO scores if response-shift effects alter the construct or meaning of PROMs.
Score calculation	Calculates score depending upon the response type of items and the availability of scoring instructions and algorithms.

**Note:** See Appendix A for the COSMIN Guideline definitions of the measurement properties and evidence used to identify psychometric properties.

## 2. Choosing the Right Measure

### 2.1. Consideration 1: Purpose of Using a PROM

The first and most important consideration in choosing a PROM is determining the purpose of using the measure: Is this measure to be used for health status classification (e.g., based on symptom severity), for prediction of future outcomes (e.g., symptom onset, late effects, survival), or for communication in clinical decision making (e.g., a starting point for patient–provider conversation)? There are many PROMs available that can serve each of these purposes. Previous systematic review studies have reported the existing PROMs used with childhood cancer survivors [22,23,24,25]. Clearly defining the purpose of using the PROM will narrow the field of available PROMs and allow clinicians to choose a PROM that is appropriate for their patient population and purpose.

### 2.2. Consideration 2: Health Profile vs. Health Preference

Health profile measures are useful for examining the impact of different cancers or treatments on specific PRO domains, and health preference measures are often used in comparative effectiveness analyses for medical decision making [26,27]. Health profile measures are created by identifying the constructs of interest, generating items, field testing the measure, and modifying items to appropriately assess the PROs [25]. In the health preference approach, the respondent’s perception of their health state is weighted by the general public’s preferences for various health states. By integrating societal preferences, this approach can be used for decision making about the distribution of health care resources and a norm-based interpretation of the intervention effects. To date, health profile PROMs (e.g., Patient-Reported Outcomes Measurement Information System (PROMIS), 36-Item Short Form Health Survey (SF-36), Minneapolis–Manchester Quality of Life Instrument (MMQL)) are more commonly used in survivorship care compared to health preference PROMs (e.g., Health Utilities Index (HUI), Short-Form Six-Dimension (SF-6D), and EQ-5D) [28,29]. However, health preference PROMs can be used as standardized means of assessing overall HRQOL, through cost–utility or comparative effectiveness analyses, for comparing the efficacy of clinical interventions and the lifelong impact of living with CHCs [30,31,32]. In the clinical case where a survivor with different treatment options for their CHCs or other medical needs may be interested in how the treatments differentially impact HRQOL so they can be part of the decision-making process, the comparative utility of health preference measures would be useful [33]. In the clinical case where the physician is interested in examining PROs of specific domains (e.g., physical, psychological) for target interventions, health profile measures would be more appropriate.

### 2.3. Consideration 3: Measurement Properties

Previous reviews [22,23,24,25,28] highlight the importance of considering the measurement properties of a PROM as a necessary precursor of clinical use because a poorly designed instrument may include items with content that is not appropriate for childhood cancer survivors [25]. The Consensus-Based Standards for the Selection of Health Measurement Instruments (COSMIN) Guidelines [34,35,36] are a framework for examining the measurement properties of PROMs, and these properties must be considered in the context of caring for survivors. Table 2 summarizes the importance of each measurement property and its relevance for clinicians, and Appendix A provides the COSMIN definitions and evidence used to identify each measurement property. Below we summarize a few key measurement properties that are particularly relevant to the clinical application of PROMs in childhood cancer survivors.

#### 2.3.1. Content Appropriateness

PROMs assess a range of content areas across physical, psychological, and social domains for pediatric cancer populations, summarized in previous review studies [22,28,37]. Common content areas are physical symptoms and emotional distress [38]. Currently, no PROMs are comprehensive in their content for this population, although the PROMIS is the most comprehensive PROM to date and may remain so until survivor-specific PROMs are developed. Additional content domains that are relevant to childhood cancer survivors but are not covered by any widely used measures include cancer-related stigma, infertility concerns, chronic symptoms, and future health expectations. Generic PROMs (e.g., PROMIS) are useful for comparing PROs across diagnoses or populations, but PROMs with items tailored to childhood cancer survivors allow clinicians to longitudinally assess differences in survivorship-specific PROs. While these tailored PROMs are not widely available yet, the Patient-Reported Outcomes Version of the Common Terminology Criteria for Adverse Events for Survivors of Childhood Cancer (PRO-CTCAE-SCC) is currently being validated for this purpose [39].

#### 2.3.2. Responsiveness

Responsiveness is the ability of a PROM to detect a change in health status over time and is valuable for assessing long-term follow-up, which is of particular importance for survivors as long-term survival rates increase [40]. Patient- and clinically-based anchors (e.g., patient-rated global improvement, clinician ratings) can be used to assess responsiveness [40,41]. If anchors are selected that are not relevant to childhood cancer survivors, then any detected change may not be meaningful. Another issue with assessing responsiveness is choosing the appropriate amount of time between PRO assessments to detect a change. One option is choosing time points triggered by a change in clinical markers or health status (e.g., onset or worsening of late effects).

#### 2.3.3. Predictive Validity

Predictive validity is another measurement property often lacking in existing PROMs used with survivors. A PROM possessing high predictive validity is clinically useful for healthcare providers to anticipate disease progression and plan for preventive interventions. Baseline PROs or the PRO change pattern (e.g., worsened pain) can be used alongside treatment and demographic variables to predict adverse health events and even premature mortality in survivors [42].

#### 2.3.4. Response-Shift Effects

Response shift is a phenomenon in which changes in a person’s responses may reflect shifts in their values, standards, or conceptualization of the PROs that are triggered by clinical interventions or a change in underlying health conditions [43]. Response-shift effects reflect a psychological adaptation during the cancer journey [44]. They may impact the empirically shown benefits of clinical interventions if survivors use different internal standards to assess PROs over time given changes in their underlying health conditions or for other reasons [45]. In examining the efficacy of an intervention with survivor-reported QOL as the outcome, clinicians want to know if improved QOL after the intervention is due to the intervention itself or survivors’ adaptation to the intervention [45]. Potential catalysts for response shift include changes in health states, worsened late effects, and moving from one developmental stage to the next [46]. Response shift in pediatric cancer survivors is still understudied, so it is unclear how the developmental age of young survivors impacts response-shift effects [47].

## 3. Ensuring Participants Complete the PROM

### 3.1. Consideration 4: Data Collection Method

Historically, PROMs have been administered via paper-and-pencil modes. These methods have transitioned to electronic platforms through advanced technologies (e.g., ePROs, mobile devices) to ease the burden of data collection on clinicians and patients [48,49]. For example, CATs take the respondents’ previous responses into account and an algorithm selects the most informative subsequent items [50]. Compared to traditional PROMs with a fixed number of items, CATs are less burdensome, more time-efficient [51], and provide higher measurement precision (i.e., lower measurement errors) [52,53].

#### mHealth and Smartphone Applications

Other ePRO platforms that may be useful for data collection include text messages and applications embedded in mobile devices (known as mHealth). Some mHealth applications used with survivors are electronic versions of original PRO surveys (e.g., Cancer Survivors Self-Efficacy Scale [54], PROMIS [55]). These applications provide opportunities for real-time assessments and notification reminders for patients to complete the survey. Some applications (e.g., Pain Squad App [56]) embed PRO measurement items within gameplay. Gamified applications provide incentives (e.g., points) for young survivors to complete PROMs, turning mundane survey completion into a fun task [56,57]. However, gamified intervention studies have shown improvement in physical and psychological outcomes [58], so gamified PROM applications may place a survivor in a different context (e.g., improved mood, distraction) from traditional PROM assessment that may alter the perception of the respondent’s genuine outcomes [59].

### 3.2. Consideration 5: Data Collection Frequency and Longitudinal Assessment

Traditional PROMs typically have a recall period of one week or longer (e.g., the Pediatric Quality of Life Inventory (PedsQL) has 7-day and 30-day options). Users need to choose the specific recall period that meets their purpose. For survivors at low risk or in a chronic phase of clinical care, a 30-day recall may be sufficient, but high-risk survivors need to respond with a shorter recall period (e.g., 7 days, 24 h). ePROMs and advanced methods (e.g., ecological momentary assessment) enable the collection of real-time PROs, with shorter recall periods (e.g., multiple times per day, if a survivor experiences excessive symptoms) between clinic visits [60]. If PRO data are collected consistently in real time, then PROs could be tracked to closely monitor disease progression and treatment side effects that can impact treatment adherence [61]. Key questions to be considered include how often can we reasonably ask survivors to report PROs? How do we factor in reporting fatigue when patients are asked to complete a PRO survey multiple times per day or week? As the population of cancer survivors increases, longitudinal PRO follow-up is critical for tracking outcomes.

PROs should ideally be collected throughout the survivorship journey, from childhood to adulthood. Most existing PROMs were developed specifically for either children or adults, but these measures are not calibrated to the same metric. For example, items used to measure physical functioning for adults and children are different, depending on the age-appropriate type of physical activity. The PROMIS is currently being calibrated to span all ages for clinical practice [62]. This line of research is ongoing and has great potential for longitudinally tracking PROs in childhood cancer survivors [47]. Key questions to be resolved in future research include what constructs are clinically meaningful to track from childhood into adulthood? How should discrepancies between age-appropriate content be handled? One potential solution is to identify common items between pediatric and adult PROMs as a bridge and then calibrate age-specific items.

### 3.3. Consideration 6: Survivor Capacity

Some patient groups at different developmental stages deserve special considerations in the design of PROMs. Survivors as young as eight years old can reliably report their PROs [63]. Children younger than eight years old need questionnaires with age-appropriate language and reading levels. Typically, children younger than five years old, without the ability to read, require a visual response scale (e.g., FPS-R [64]), interviewer-read items, or a proxy report (usually a parent/caregiver). Childhood cancer survivors at any age left with functional deficits from cancer therapy may require special assistance for PRO assessment [65,66]. In common practice, there are two approaches to assessing the cognitive capacity of survivors: asking caregivers if their children can self-report PROs or using standard cognitive assessment batteries in some medical centers. Similar to the younger age group, visual response scales or interviewer-read items may be used to collect PROs from survivors with cognitive deficits. Hearing and vision loss from cancer or treatment can affect a survivor’s ability to self-report PROs [67,68]. Therefore, it is critical to design separate modules of PROMs for survivors who require special assistance for completing PRO assessment (e.g., interactive voice response systems for survivors with vision loss [69], interpreters or hearing-assistive technology for survivors with hearing loss [70]) and characterize their measurement properties.

### 3.4. Consideration 7: Self- vs. Proxy Reports

There is general agreement that PROs should be collected directly from survivors [41,71]. Due to the limited capability of some survivors to complete PROs, and the role of caregivers in shared decision making for healthcare, caregiver proxy reports may be considered. There is evidence that caregivers tend to overestimate poor PROs experienced by young survivors compared to self-reports from children and adolescents [72]. However, this difference may not be as significant in very young children, suggesting that proxy reports are appropriate for that age group [73]. Additionally, some subjective constructs lend themselves to self-reporting more easily than others, such as symptoms (e.g., pain, fatigue) and psychological and social functioning [74]. From the measurement perspective, it is acceptable to use either self- or proxy reports for a specific PROM as long as there is evidence of measurement invariance between PROs rated by the survivor and the caregiver proxy [75]. If measurement non-invariance exists, it is important to integrate the dyadic data of pediatric PROs collected from survivors and proxies to facilitate clinical research (e.g., cost–utility analysis for different therapeutic options); however, a methodological framework for integrating dyadic scores is still understudied.

## 4. Interpreting the Results

### 4.1. Consideration 8: Scoring Methods

Most existing PROMs employ a summation or mean scoring method with equal weight on all items to calculate domain scores. Recent PROMs (e.g., PROMIS) use more advanced methods (e.g., IRT, CFA) by taking into account possible differences in the relationship between the construct of interest and individual items (i.e., unequal weights on different items) for calculating the domain scores [76,77,78]. Weighted methods may be more sensitive to changes in health conditions over time compared to unweighted methods, providing evidence of responsiveness for tracking PROs in long-term survivors [79]. Although PRO scores derived from summation methods may not accurately inform outcome differences compared to those from more advanced scoring methods, unweighted scoring methods are easier to use than weighted methods [76,78].

### 4.2. Consideration 9: Clinical Meaning and Interpretability

Two important, often missing rubrics for PROMs are established, clinically meaningful severity cutpoints and MIDs to facilitate the interpretability of PRO scores. Clinicians have reported the lack of clinically meaningful cutpoints and MIDs as a major barrier to using PROMs in practice [17,18,19]. Cutpoints are used to interpret the severity of PRO scores for medical decision making, whereas MIDs are used for examining intraindividual PRO changes over time that are meaningful to the patient or clinician [40,80]. There may be multiple MIDs for each PRO for different populations [40]. Various anchor-based methods, such as patient-centered anchors (i.e., patients rate their health changes at different time points in a clinical trial, typically through a global rating of change scale) and clinically-based anchors (e.g., change in several nausea episodes, change in cognitive task performance), are used as the primary approach to establish MIDs for the clinical interpretation of PROMs [41]. Currently, cutpoints and MIDs developed for many PROMs do not rigorously apply clinically based, survivorship-relevant anchors, which limits their clinical utility in pediatric cancer survivors. The PROMIS website provides useful guidelines to facilitate meaningful interpretations of severity and change of PRO scores including cutpoints and MIDs, but clinicians should keep in mind that these scores were not developed specifically for childhood cancer survivors [81]. Clinicians could consider using novel methods such as scale judgment or bookmarking, described on the PROMIS website [81], as well as survivorship-sensitive clinical anchors (e.g., treatment modalities) alongside patient-centered anchors [38], to develop cutpoints and MIDs for childhood cancer survivors to facilitate score interpretation.

## 5. Selecting a Strategy for Clinical Response

### Consideration 10: Integration into the Clinical Workflow

Efficiently collecting PRO data from survivors at the right time and in the right place and seamlessly integrating these data into the clinical workflow are key strategies to improve PRO implementation. In high-income countries where there is existing infrastructure for digitally summarizing clinical data in hospitals, the EHR could be ideally used to collect and summarize PRO data. Though clinicians have expressed a preference for PRO data to be integrated into their existing EHRs rather than into a separate system, there are numerous disparate EHR systems across healthcare systems that have different technological capabilities or limitations [16]. Systematically collecting PRO data has not become standardized clinical practice, but some hospitals have overcome many barriers and incorporated PRO data into EHRs and survivorship care [82]. Recommendations for integrating PROMs into the EHR for adult cancer patients have been discussed, but the EHR integration for survivorship care that requires data sharing between oncologists and primary care physicians is lacking [83]. If PRO data are integrated into existing hospitals’ EHRs, then clinical teams can easily track longitudinal PROs and identify survivors who need services over the survivorship care trajectory. One example of an integration framework that uses a novel PRO data collection methodology is described below (see Figure 1). By using the framework displayed in Figure 1, the burden of collecting, transcribing, analyzing, and interpreting PROMs is taken off the clinical staff and put on the EHR and technological infrastructure to practically facilitate clinical implementation.

## 6. Illustration for Integrating Novel PRO Data Collection into the Clinical Workflow

The platforms that house both the PROM data collection and results must be user-friendly for both patients and clinicians. The first step in Figure 1 illustrates that PRO data can be collected via the survivor’s personal device (e.g., smartphone, tablet, computer). The types of PRO data collected at this step could be traditional PROMs, patient-generated health data (PGHD), or novel data collection methods such as unstructured PRO data analyzed with natural language processing (NLP) and machine learning (ML) (elaborations for PGHD, NLP, and ML to follow).

### 6.1. Patient-Generated Health Data

There is a growing interest in assessing PROs alongside biometrics and behavioral tracking (known as PGHD) for clinical use [84]. Biometrics and behavioral data (e.g., skin temperature, heart rate variability, respiratory rate, sleep behavior, activity steps) are typically collected via wearable trackers or mHealth devices (e.g., pedometers, Apple watches, smartphones) [85]. PGHD can be collected momentarily to reflect real-time health status, but this approach results in large amounts of data. AI can be used to analyze these data [86]. Both biometrics and PROs can be integrated into EHRs to obtain a more robust picture of health outcomes for survivors with the progression of late effects or anticipation of CHCs.

### 6.2. Novel Unstructured PRO Data Collection

Collecting and assessing PROs is challenging in busy clinics. An alternative method is a reliance on conversations between patients and clinicians during clinical encounters and subsequent content annotations [87]. This method for collecting PROs is less structured than conventional PROMs but allows patients to express their experiences freely rather than being confined to the content of the standardized PROM items. It is important to use conversation guidelines to ensure clinicians are asking appropriate questions that elicit responses from patients about their PROs. Another source of free-text PRO data is historical medical notes [88,89,90]. These unstructured PRO data alongside other clinical parameters collected during cancer diagnosis or follow-up visits may be useful for developing a prognostic model for predicting late effects [91].

### 6.3. Data Repository and Analysis

All PRO data (i.e., structured, unstructured), other PGHD, and clinical data related to PROs would be downloaded to an informatics repository (step 2, Figure 1). Here, the data are cleaned and prepared for analysis. Currently, collecting PROs and other PGHD and integrating them into EHRs is not routinely performed in pediatric oncology [92]. One of the barriers is the lack of information technology support for transferring the data among different mHealth devices to EHRs due to poor interoperability [86]. Once the data are in the repository, they can be analyzed (step 3, Figure 1). Analyzing unstructured PRO data requires novel pipelines and methods. One approach is NLP, which uses AI to extract and annotate unstructured PRO data and identify features, such as specific types and attributes of symptoms [93]. ML, typically neural network techniques, is useful for examining associations of PRO features with clinical outcomes [94]. These novel approaches could be integrated into hospital EHRs and used by clinicians to predict disease progression, adverse events, and premature mortality. In these prediction models, further research is warranted to determine clinically meaningful NLP/ML-derived features to denote complex PRO issues for childhood cancer survivors. It is also important to compare measurement properties between NLP/ML-based methods with traditional survey-based methods and identify clinical barriers of implementing these AI-driven methods.

### 6.4. Integration into the EHR and Patient Portal

Once the data are analyzed, the results can be uploaded to the clinical EHR (step 4, Figure 1) and/or the survivor’s patient portal (step 6, Figure 1). From the clinical EHR, providers can track longitudinal PROs. It is critical to design a platform to facilitate the systematic collection of longitudinal PROs, from childhood to adulthood, among pediatric cancer populations, and to share the data among all healthcare providers throughout the cancer journey. PRO results need to be displayed in the clinical EHR in a way that is easily understandable and clinically actionable (e.g., graphs of PROM scores over time, highlighted scores that reach cutpoints or MID thresholds). In the case-management platform (step 5, Figure 1), clinicians can be alerted to survivors whose scores are above the threshold and warrant changes in treatment or medical intervention.

PROMs integrated into the EHR can facilitate communication with survivors, while also providing automated recommendations for self-care strategies directly to survivors (step 6, Figure 1). The ePRO platforms embedded in patient portals may include automatic reminders for regular PRO assessment, automatic score calculation with built-in algorithms on the backend, rapid score interpretation through referring to clinical evidence, and recommendations for disease management.

## 7. Conclusions and Future Directions

PROs in childhood cancer survivors are vital for monitoring the late effects of treatment and guiding long-term cancer survivorship care. While there are time, resource, and interpretation barriers that clinicians report in integrating PROMs into their practice, we provide 10 considerations for clinicians to overcome these implementation barriers. Table 1 can be used as a checklist to remind clinicians, when implementing PROMs into their practice, how to ensure appropriate PRO assessment by considering key factors such as the purpose of PRO assessment, the type of PROMs, the characteristics of the target population, the data collection method, and score interpretations. One organization that facilitates PROM implementation is the International Consortium for Health Outcomes Measurement, which was founded in 2012 to develop core sets of PROMs to be implemented in clinical care [95]. They advocate for standardized, core sets of PROMs to support the integration of the patient voice across value-based healthcare and help with implementation for clinicians. Currently, they do not have any PROM core sets for childhood cancer survivors, which should be addressed in future research. However, the PROMIS and other PROMs have been used successfully to evaluate a variety of PROs in childhood cancer survivors [22,24,96,97].

Childhood cancer survivors’ clinicians report receiving limited training or a lack of understanding of PRO interpretation, which inhibits PRO implementation [17,18,19]. Professional pediatric societies could promote the use of PROMs through education and training in implementing PROMs in clinical survivorship settings. Incorporation of specific types (e.g., symptoms by the anatomic site) and attributes (e.g., symptom presence, severity, interference) of PROs into survivorship care may be beneficial. The Children’s Oncology Group Long-Term Follow-Up Guidelines for Survivors of Childhood, Adolescent, and Young Adult Cancers [98] conventionally use treatment modalities to stratify patients at risk of developing late effects [99,100]. General practitioners may not be aware of a survivor’s pediatric cancer treatment information, and impaired PROs are associated with progressive CHCs and premature mortality [14,101], so in survivorship care, regularly assessed PROs could be included as additional stratification factors for identifying patients at risk for late effects. In some low- and middle-income countries, one barrier to integrating PRO data collection into the clinical workflow is the lack of EHR infrastructures and technological training. There was a recent call for high-income countries to partner with low- and middle-income countries in developing digital health platforms to improve global healthcare [102] and incorporating PROs into these systems could be an effective way to foster patient-centered survivorship care in these settings.

The future application of PROMs will be facilitated by technological and methodological advances currently underway in health research. As PROMs continue to improve and more clinicians implement these measures, we will gain a stronger understanding of the experiences of childhood cancer survivors and provide better care as more pediatric cancer patients become long-term cancer survivors.

## Figures and Tables

**Figure 1 cancers-15-01024-f001:**
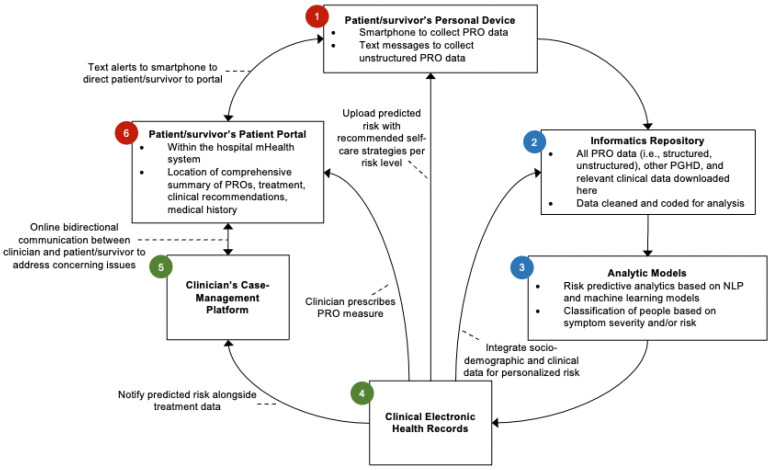
Flow for integrating PRO data collection into clinical workflow. The red colors represent patient-facing steps, the blue colors represent informatics/structure, and the green colors represent clinician-facing steps. Double-headed arrows indicate bidirectional interconnections between the two platforms (e.g., two-way patient–provider communication).

## Data Availability

No new data were generated or analyzed in support of this research.

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
