# Peer review of "Ten Considerations for Integrating Patient-Reported Outcomes into Clinical Care for Childhood Cancer Survivors"

_cancers, 2023, doi:10.3390/cancers15041024_

Round 1

Reviewer 1 Report

Thank you for the opportunity to review the manuscript entitled “Ten considerations for integrating patient-reported outcomes (PROM) into clinical care for childhood cancer survivors”. The topic is important as in the adult oncology, it has been stated that taking into consideration PROMs, may even have effect on the survival of the patient.

As there are still barriers to integrating PROMs into clinical practice, such as constraints in clinical validity, meaningful interpretation, and technology-enabled administration of the measures, this article discusses these, and proposes 10 important considerations for appropriate PROM integration into clinical care with special focus on childhood cancer survivors.

This is the first manuscript profoundly describing the ideology in choosing suitable PROMs specifically for the childhood cancer survivors. It gives very valuable and comprehensive information on the topic in a way that is understandable for clinicians on the field of pediatric oncology. It is also important to get more input on finding a reliable way of including the psychosocial well-being/risks of the survivor into the 3-level late-effect risk assessment tool which at the moment takes into account only the disease and treatment factors with cumulative exposures but not the mental part.

General comments

The manuscript is clearly structured and very relevant for the field. It describes the whole PROM-concept in a compact way.

The structure of the 10 given considerations includes quite a long referring to COSMIN guidelines (points 2.3-2.5) about the measurement properties but I feel that this is important part of the review and gives good examples of useful tools having good properties when considering from the point of view of cancer survivorship.

On row 186, it is stated that currently, cut-points and minimally important differences (MIDs) developed for many PROMs do not rigorously apply clinically-based, survivorship-relevant anchors, which limits their clinical utility in pediatric cancer survivors. However, there are a couple of examples where the PROMIS-tool is found useful in evaluation of survivors. These references might be worth considering:

DeWalt DA, et al. PROMIS ® pediatric self-report scales distinguish subgroups of children within and across six common pediatric chronic health conditions. Quality of life research, 2015, Vol.24 (9), p.2195-2208

Sopfe J, et al. Evaluation of the v2.0 Brief Profiles for Sexual Function and Satisfaction PROMIS in Adolescent and Young Adult Childhood Cancer Survivors. Journal of adolescent and young adult oncology, 2021, Vol.10 (4), p.418-424

There is also very good discussion about modern methods of data collection (row 198 onwards) as well as about the data collection frequency and the importance of finding PROMs that are suitable from childhood to adulthood and enable/have included a property of comparable scaling and scale calibration such that comparable results can be obtained at all ages. This enables a real follow-up of the child patient becoming an adult.

It is important to be able to include PROMs into clinical workflow as described at point 2.10. and figure 1.

A merit should be given also to point 3.3 (row 319 onwards) as it must be the clear aim in the future to use all data that are collected in daily routines and develop/implement machine learning methods for analyzing electronic medical records in order to find out early signs of impaired psychosocial health, and, furthermore, offer interventions based on these warning signs.

The tables and figures are appropriate.

The reference list seems to be appropriate. If the authors could add the few examples of publications on childhood cancer survivors evaluated successfully with the PROMIS-tools, it might add something promising after the statement about ICHOM (“Currently, they do not have any PROM core sets for childhood cancer survivors, which should be addressed in future research”)

Author Response

Reviewer 1

Thank you for the opportunity to review the manuscript entitled “Ten considerations for integrating patient-reported outcomes (PROM) into clinical care for childhood cancer survivors”. The topic is important as in the adult oncology, it has been stated that taking into consideration PROMs, may even have effect on the survival of the patient.

As there are still barriers to integrating PROMs into clinical practice, such as constraints in clinical validity, meaningful interpretation, and technology-enabled administration of the measures, this article discusses these, and proposes 10 important considerations for appropriate PROM integration into clinical care with special focus on childhood cancer survivors.

This is the first manuscript profoundly describing the ideology in choosing suitable PROMs specifically for the childhood cancer survivors. It gives very valuable and comprehensive information on the topic in a way that is understandable for clinicians on the field of pediatric oncology. It is also important to get more input on finding a reliable way of including the psychosocial well-being/risks of the survivor into the 3-level late-effect risk assessment tool which at the moment takes into account only the disease and treatment factors with cumulative exposures but not the mental part.

General comments

  1. The manuscript is clearly structured and very relevant for the field. It describes the whole PROM-concept in a compact way.

The structure of the 10 given considerations includes quite a long referring to COSMIN guidelines (points 2.3-2.5) about the measurement properties but I feel that this is important part of the review and gives good examples of useful tools having good properties when considering from the point of view of cancer survivorship.

Response: We appreciate this reviewer’s comments on the structure of the manuscript.

  1. On row 186, it is stated that currently, cut-points and minimally important differences (MIDs) developed for many PROMs do not rigorously apply clinically-based, survivorship-relevant anchors, which limits their clinical utility in pediatric cancer survivors. However, there are a couple of examples where the PROMIS-tool is found useful in evaluation of survivors. These references might be worth considering:

DeWalt DA, et al. PROMIS ® pediatric self-report scales distinguish subgroups of children within and across six common pediatric chronic health conditions. Quality of life research, 2015, Vol.24 (9), p.2195-2208

Sopfe J, et al. Evaluation of the v2.0 Brief Profiles for Sexual Function and Satisfaction PROMIS in Adolescent and Young Adult Childhood Cancer Survivors. Journal of adolescent and young adult oncology, 2021, Vol.10 (4), p.418-424

Response: Thank you for this useful comment. To address the availability of cutpoints and MIDs for the PROMIS measures, we have added the following sentence to page 9, which now reads: “The PROMIS website provides useful guidelines to facilitate meaningful interpretations of severity and change of PRO scores including cutpoints and MIDs, but clinicians should keep in mind that these scores were not developed specifically for childhood cancer survivors [81]. Clinicians could consider using novel methods such as scale judgment or bookmarking, described on the PROMIS website [81], as well as survivorship-sensitive clinical anchors (e.g. treatment modalities) alongside patient-centered anchors [38], to develop cutpoints and MIDs for childhood cancer survivors to facilitate score interpretation.”

  1. There is also very good discussion about modern methods of data collection (row 198 onwards) as well as about the data collection frequency and the importance of finding PROMs that are suitable from childhood to adulthood and enable/have included a property of comparable scaling and scale calibration such that comparable results can be obtained at all ages. This enables a real follow-up of the child patient becoming an adult.

Response: Thank you for this positive feedback on the discussion of data collection in the manuscript.

  1. It is important to be able to include PROMs into clinical workflow as described at point 2.10. and figure 1.

Response: We appreciate this reviewer’s comment and are glad that our focus on the integration of PROMs into the clinical workflow was clear and useful.

  1. A merit should be given also to point 3.3 (row 319 onwards) as it must be the clear aim in the future to use all data that are collected in daily routines and develop/implement machine learning methods for analyzing electronic medical records in order to find out early signs of impaired psychosocial health, and, furthermore, offer interventions based on these warning signs.

Response: Thank you to the reviewer for providing this merit.

  1. The tables and figures are appropriate.

Response: Thank you for this positive comment about the tables and figures.

  1. The reference list seems to be appropriate. If the authors could add the few examples of publications on childhood cancer survivors evaluated successfully with the PROMIS-tools, it might add something promising after the statement about ICHOM (“Currently, they do not have any PROM core sets for childhood cancer survivors, which should be addressed in future research”)

Response: Thank you for this helpful suggestion. We have added the following statement to page 12, which now reads: “However, the PROMIS and other PROMs have been used successfully to evaluate a variety of PROs in childhood cancer survivors [22,24,96,97].”

Reviewer 2 Report

Thank you for the invitation to review this well-written manuscript on the use of PRO/PROMs in the clinical care of childhood cancer survivors. Increasingly the use of PRO/PROMs facilitates the cross-study comparison in research and produces standardized measures for common psychosocial and physical health outcomes among cancer patients and care partners. In the ten practical recommendations for the integration of PRO/PROMs into the clinical care of childhood cancer survivors this manuscript acts as a "hand book" if you will, for what the future integration of PRO/PROMs in pediatric cancer care could look like. I have some minor recommendations below.

Minor comments: 

Section 2.3.4. - How does the developmental age/stage of pediatric, adolescent, young adult survivors also influence response-shift effects? This seems an important, unique, possibility to consider for this age of cancer survivors.

Section 2.8 - I'm not sure the content on IQ is helpful as it seems quite impractical to evaluate IQ in a clinical setting when administering PROs. 

 Table 1: I suggest avoiding the "pro/con" terminology as the acronyms PRO and PROM are so commonly used and may be easily conflated with "pro" in "pros/cons". 

Table 1: Ensure all acronyms are defined at first use (e.g., CFA, IRT, CHIP, FPSR, ChIMES etc.)

  •  

Author Response

Reviewer 2

Thank you for the invitation to review this well-written manuscript on the use of PRO/PROMs in the clinical care of childhood cancer survivors. Increasingly the use of PRO/PROMs facilitates the cross-study comparison in research and produces standardized measures for common psychosocial and physical health outcomes among cancer patients and care partners. In the ten practical recommendations for the integration of PRO/PROMs into the clinical care of childhood cancer survivors this manuscript acts as a "hand book" if you will, for what the future integration of PRO/PROMs in pediatric cancer care could look like. I have some minor recommendations below.

Minor comments: 

  1. Section 2.3.4. - How does the developmental age/stage of pediatric, adolescent, young adult survivors also influence response-shift effects? This seems an important, unique, possibility to consider for this age of cancer survivors.

Response: Thank you for this well-taken point regarding the response shift associated with the developmental stage of the survivors. We have added the sentences to page 7, which now reads: “Potential catalysts for response shift include changes in health states, worsened late effects, and moving from one developmental stage to the next [46]. Response shift in pediatric cancer survivors is still understudied, so it is unclear how the developmental age of young survivors impacts response-shift effects [47].”

  1. Section 2.8 - I'm not sure the content on IQ is helpful as it seems quite impractical to evaluate IQ in a clinical setting when administering PROs. 

Response: Thank you for this useful comment. We agree with this reviewer, and have edited this section on page 8, which now reads: “In common practice, there are two approaches to assessing the cognitive capacity of survivors: asking caregivers if their children are able to self-report PROs or using standard cognitive assessment batteries in some medical centers.”

  1. Table 1: I suggest avoiding the "pro/con" terminology as the acronyms PRO and PROM are so commonly used and may be easily conflated with "pro" in "pros/cons". 

Response: Thank you for this helpful comment. We have edited the pros/cons language to advantages/disadvantages.

  1. Table 1: Ensure all acronyms are defined at first use (e.g., CFA, IRT, CHIP, FPSR, ChIMES etc.)

Response: We have edited the manuscript to define all acronyms at first use.

Reviewer 3 Report

Thank you for the opportunity to review this manuscript which proposes 10 considerations for integrating Patient Reported Outcome Measures (PROMS) in the clinical care of childhood cancer survivors. This is a timely and important topic, as PROs are increasingly being collected in both research and clinical care, but they have had less uptake in the pediatric cancer survivorship population. While the authors have some helpful suggestions for integrating PROMs in cancer survivorship, as written the manuscript feels more targeted towards PROMS researchers, and others with a great deal of familiarity with PROMs, than towards clinicians who may be struggling to include PROMs in their clinical practice. For example, the language used throughout includes jargon familiar to PROMs researchers, but which may be less familiar to those who might be interested in including PROMs in their clinical practice. Therefore, while this manuscript explores an important topic, as currently framed, it is not as helpful as it could be for its apparent target audience. It would be strengthened by consideration of the following:

Introduction: The introduction currently includes a great deal of background on PROs across cancer care. It would be strengthened by narrowing of focus on PROMs in pediatric cancer and pediatric cancer survivorship, as well as discussing the clinical utility and challenges of clinical (as opposed to research) collection of PROMs (further expanding paragraphs 3 & 4).

10 Suggestions: As written, the 10 suggestions include a lot of information about PROMs, but don’t fully distill the information in a way that would help a busy clinical provider understand how to incorporate PROMs in their clinical practice. For example, Consideration 5, “Clinical Meaning and interpretability” includes interesting background on the difference between minimally important differences and meaningful severity cut points. However, there are no suggestions about how to use this background to help select appropriate cut points in clinical practice.

The whole piece may be strengthened by the use of a clinical example, and a slight rearrangement of the 10 suggestions in an order and grouping that might support a clinician in implementing PROMs in their practice. In this way, they could help clinicians to phrase and frame their clinical question, then group the 10 suggestions into considerations for selecting the right measures (currently 1, 2, 3), ensuring that participants complete the measures (6, 7, 8, 9), interpreting the results (4 & 5), selecting a strategy for clinical response (10 - though this section should be expanded).

Tables and Figures:

Table 1 presumes a high baseline knowledge of PROMs and associated jargon which though described in more detail in the text, limits its value as a standalone table.

Figure 1 is extremely confusing. Perhaps color coding patient facing considerations and Clinician/HER considerations may help, but the arrows pointing in multiple directions, and the out of sequence numbering makes it very challenging to understand where to start and proceed.

Author Response

Reviewer 3

Thank you for the opportunity to review this manuscript which proposes 10 considerations for integrating Patient Reported Outcome Measures (PROMS) in the clinical care of childhood cancer survivors. This is a timely and important topic, as PROs are increasingly being collected in both research and clinical care, but they have had less uptake in the pediatric cancer survivorship population. While the authors have some helpful suggestions for integrating PROMs in cancer survivorship, as written the manuscript feels more targeted towards PROMS researchers, and others with a great deal of familiarity with PROMs, than towards clinicians who may be struggling to include PROMs in their clinical practice. For example, the language used throughout includes jargon familiar to PROMs researchers, but which may be less familiar to those who might be interested in including PROMs in their clinical practice. Therefore, while this manuscript explores an important topic, as currently framed, it is not as helpful as it could be for its apparent target audience. It would be strengthened by consideration of the following:

  1. Introduction: The introduction currently includes a great deal of background on PROs across cancer care. It would be strengthened by narrowing of focus on PROMs in pediatric cancer and pediatric cancer survivorship, as well as discussing the clinical utility and challenges of clinical (as opposed to research) collection of PROMs (further expanding paragraphs 3 & 4).

Response: Per this reviewer’s suggestion, we have revised some statements in the Introduction section by removing some research-centric languages and focusing on the clinical utility and barriers rather than the barriers to using PROs in research.

  1. 10 Suggestions: As written, the 10 suggestions include a lot of information about PROMs, but don’t fully distill the information in a way that would help a busy clinical provider understand how to incorporate PROMs in their clinical practice. For example, Consideration 5, “Clinical Meaning and interpretability” includes interesting background on the difference between minimally important differences and meaningful severity cut points. However, there are no suggestions about how to use this background to help select appropriate cut points in clinical practice.

Response: We appreciate this useful feedback on the important concept of incorporating PROMs into the busy clinic. We have added some practical recommendations to the section regarding cutpoints on page 9, which now reads: “Clinicians could consider using novel methods such as scale judgment or bookmarking, described on the PROMIS website [81], as well as survivorship-sensitive clinical anchors (e.g., treatment modalities) alongside patient-centered anchors [38], to develop cutpoints and MIDs for childhood cancer survivors to facilitate score interpretation.” We have also emphasized the clinical utility of the proposed framework (Figure 1) to integrate PROs into EHRs on page 10, which now reads: “By using the framework displayed in Figure 1, the burden of collecting, transcribing, analyzing, and interpreting PROMs is taken off the clinical staff and put on the EHR and technological infrastructure to practically facilitate clinical implementation.”

  1. The whole piece may be strengthened by the use of a clinical example, and a slight rearrangement of the 10 suggestions in an order and grouping that might support a clinician in implementing PROMs in their practice. In this way, they could help clinicians to phrase and frame their clinical question, then group the 10 suggestions into considerations for selecting the right measures (currently 1, 2, 3), ensuring that participants complete the measures (6, 7, 8, 9), interpreting the results (4 & 5), selecting a strategy for clinical response (10 - though this section should be expanded).

Response: We agree that rearranging the 10 recommendations would strengthen the flow of this paper and have adjusted the manuscript accordingly. We also added a clarification at the end of the Introduction section on page 2, which now reads: “This article provides 10 clinically actionable considerations for clinicians using PROMs for selecting the right measures, ensuring survivors complete the measures, interpreting the results, and selecting a strategy for clinical response (see Table 1).” Additionally, we added a clinical example to the health profile vs. health preference section on page 5, which now reads: “In the clinical case where a survivor with different treatment options for their CHCs or other medical needs may be interested in how the treatments differentially impact HRQOL so they can be part of the decision-making process, the comparative utility of health preference measures would be useful [33]. In the clinical case where the physician is interested in examining PROs of specific domains (e.g., physical, psychological) for target interventions, health profile measures would be more appropriate.”

  1. Tables and Figures:

Table 1 presumes a high baseline knowledge of PROMs and associated jargon which though described in more detail in the text, limits its value as a standalone table.

Response: We have defined all abbreviations and clarified/removed technical jargon in Table 1, and added the following statement to the conclusions on page 12 to state our purpose for including Table 1: “Table 1 can be used as a checklist to remind clinicians, when implementing PROMs into their practice, how to ensure appropriate PRO assessment by considering key factors such as the purpose of PRO assessment, the type of PROMs, the characteristics of the target population, the data collection method, and score interpretations.”  

  1. Figure 1 is extremely confusing. Perhaps color coding patient facing considerations and Clinician/HER considerations may help, but the arrows pointing in multiple directions, and the out of sequence numbering makes it very challenging to understand where to start and proceed.

Response: We have edited Figure 1 to represent the clinical workflow more clearly. We have also color-coded the steps for patient-facing steps, informatics/structure, and clinician-facing steps. We explained this color-coding as well as the bidirectional arrows in the figure caption.

Reviewer 4 Report

The article is relevant since it contributes to facilitating long-term follow-up for health professionals of childhood and adolescent cancer survivors.

I only add some minor comments. In table 1:

Consideration 2: Health profile vs. health preference approaches

• Health profile measures:

 Pros: useful for examining the impact of disease or treatment on specific PRO do-mains, suitable for making care decisions for individuals

 Cons: scale values and summary scores may be unfamiliar or difficult to interpret

• Health utility measures:

Pros: useful for making decisions regarding the distribution of health care resources across different diseases; common metric (0 = dead, 1 = optimal health)

Cons: preference weights developed from the general population in Common-wealth countries and may not be applicable for all countries; some scaling methods are difficult for children to complete

I suggest don´t put bullet point before Pros and Cons or you could use another kind  of bullet point

Author Response

Reviewer 4

The article is relevant since it contributes to facilitating long-term follow-up for health professionals of childhood and adolescent cancer survivors.  I only add some minor comments.

  1. In table 1: Consideration 2: Health profile vs. health preference approaches
  • Health profile measures:

Pros: useful for examining the impact of disease or treatment on specific PRO do-mains, suitable for making care decisions for individuals

Cons: scale values and summary scores may be unfamiliar or difficult to interpret

  • Health utility measures:

Pros: useful for making decisions regarding the distribution of health care resources across different diseases; common metric (0 = dead, 1 = optimal health)

Cons: preference weights developed from the general population in Common-wealth countries and may not be applicable for all countries; some scaling methods are difficult for children to complete

I suggest don´t put bullet point before Pros and Cons or you could use another kind of bullet point.

Response: Thank you for this useful suggestion. We have edited the bullet points in Table 1 to clarify that the pros and cons (edited to advantages and disadvantages) are subsumed under the “Health profile measures” and “Health utility measures” headings.

Reviewer 5 Report

Very interesting work. Some minor revisions:

- one of the big barriers, little discussed, concerns the technological training of the population and hospital equipment. In many countries both are very scarce

- for correct use in clinical practice, guidelines would be needed

- an important work was not reviewed:

Di Maio M, Basch E, Denis F, Fallowfield LJ, Ganz PA, Howell D, Kowalski C, Perrone F, Stover AM, Sundaresan P, Warrington L, Zhang L, Apostolidis K, Freeman-Daily J, Ripamonti CI, Santini D; ESMO Guidelines Committee. Electronic address: clinicalguidelines@esmo.org. The role of patient-reported outcome measures in the continuum of cancer clinical care: ESMO Clinical Practice Guideline. Ann Oncol. 2022 Sep;33(9):878-892. doi: 10.1016/j.annonc.2022.04.007. Epub 2022 Apr 21. PMID: 35462007.

Author Response

Reviewer 5

Very interesting work. Some minor revisions:

  1. one of the big barriers, little discussed, concerns the technological training of the population and hospital equipment. In many countries both are very scarce

Response: We appreciate this reviewer’s comment about the scarcity of technological training in many countries. We have added the following sentence to Consideration 10 (Integration into the clinical workflow) on page 9, which now reads: “In high-income countries where there is existing infrastructure for digitally summarizing clinical data in the hospitals, the EHR could be ideally used to collect and summarize PRO data.” We have also added the following statement to the conclusions section on page 12, which now reads: “In some low- and middle-income countries, one barrier to integrating PRO data collection into the clinical workflow is the lack of EHR infrastructures and technological training. There was a recent call for high-income countries to partner with low- and middle-income countries in developing digital health platforms to improve global healthcare [102], and incorporating PROs into these systems could be an effective way to foster patient-centered survivorship care in these settings.”

  1. for correct use in clinical practice, guidelines would be needed

Response: We agree with this reviewer and have provided references to two guidelines (from the International Society for Quality of Life Research [20] and the Patient-Centered Outcomes Research Institute [21]) in the Introduction section on page 2. We also added the statement on page 12: “Table 1 can be used as a checklist to remind clinicians, when implementing PROMs into their practice, how to ensure appropriate PRO assessment by considering key factors such as the purpose of PRO assessment, the type of PROMs, the characteristics of the target population, the data collection method, and score interpretations.”

  1. an important work was not reviewed:

Di Maio M, Basch E, Denis F, Fallowfield LJ, Ganz PA, Howell D, Kowalski C, Perrone F, Stover AM, Sundaresan P, Warrington L, Zhang L, Apostolidis K, Freeman-Daily J, Ripamonti CI, Santini D; ESMO Guidelines Committee. Electronic address: clinicalguidelines@esmo.org. The role of patient-reported outcome measures in the continuum of cancer clinical care: ESMO Clinical Practice Guideline. Ann Oncol. 2022 Sep;33(9):878-892. doi: 10.1016/j.annonc.2022.04.007. Epub 2022 Apr 21. PMID: 35462007.

Response: Thank you for recommending this article. We have incorporated the reference into the following sentence on pages 9 and 10, which now reads: “Recommendations for integrating PROMs into the EHR for adult cancer patients have been discussed, but the EHR integration for survivorship care that requires data sharing between oncologists and primary care physicians is lacking [83].”

Round 2

Reviewer 3 Report

This paper was greatly strengthened by the attached revisions. No further concerns.